# The Behavior of Two Types of Upper Removable Retainers—Our Clinical Experience

**DOI:** 10.3390/children7120295

**Published:** 2020-12-16

**Authors:** Luminita Ligia Vaida, Eugen Silviu Bud, Liliana Gabriela Halitchi, Simona Cavalu, Bianca Ioana Todor, Bianca Maria Negrutiu, Abel Emanuel Moca, Florian Dorel Bodog

**Affiliations:** 1Department of Dentistry, Faculty of Medicine and Pharmacy, University of Oradea, 1 Universitatii Str., 410087 Oradea, Romania; ligia_vaida@yahoo.com (L.L.V.); biancaioana.todor@gmail.com (B.I.T.); biancastanis@yahoo.com (B.M.N.); abelmoca@yahoo.com (A.E.M.); 2Department of Orthodontics, University of Medicine and Pharmacy Science and Technology G.E Palade, 38 Gh. Marinescu Str., 540139 Targu Mures, Romania; 3Department of Clinical Disciplines, Faculty of Dentistry, Apollonia University of Iasi, 2 Muzicii Str., 700399 Iasi, Romania; 4Department of Preclinical Sciences, Faculty of Medicine and Pharmacy, University of Oradea, 1 Universitatii Str., 410087 Oradea, Romania; simona.cavalu@gmail.com; 5Department of Surgery, Faculty of Medicine and Pharmacy, University of Oradea, 1 Universitatii Str., 410087 Oradea, Romania; fbodog@gmail.com

**Keywords:** Hawley retainer, vacuum-formed retainer, relapse, orthodontic treatment, children and adolescents

## Abstract

The Hawley retainer (HR) and the vacuum-formed retainer (VFR) are the most common removable retainers in orthodontic treatments. The aim of this retrospective study was to comparatively analyze the behavior of two types of removable retainers—HRs and VFRs—in terms of retainer damage, loss, and the rate of installation of mild or severe relapse that required recourse to certain therapeutic interventions. The study was performed on 618 orthodontic patients aged 11–17 years, average age 13.98 ± 1.51, out of which 57% were patients having VFRs and the remaining 43% having HRs in the upper arch. We performed an analysis of the two groups of patients—HRs group and VFRs group—at 6 months (T1) and at 12 months (T2) after the application of the retainer. The results showed that 6% of all the retainers were damaged, mostly at T2 (54.1%). Seven percent of all the retainers were lost, mostly at T1 (58.1%). Of all the patients, 9.1% presented mild relapse, mostly at T1 (58.9%), while 2.6% presented severe relapse. The VFRs were significantly more frequently associated with the occurrence of damage than the HRs (*p* < 0.001). Severe relapse was more frequently associated with the HRs rather than with VFRs (*p* < 0.05).

## 1. Introduction

The aim of the orthodontic treatment is not only to improve oral functioning and health but also to enhance dento-facial esthetics, self-esteem, and oral health-related quality of life [1]. Retention is a sine-qua-non condition of orthodontic treatment and its aim is to maintain the teeth in the correct position obtained at the end of the active phase of the orthodontic treatment. The omission of the retention phase has as consequence the relapse, the tendency for the initial malocclusion to reappear [2,3]. Relapse may occur due to imbalances in the gingival fibers, periodontal tissues, occlusion, or facial soft tissues [4,5]. Unwanted changes may also occur after the removal of the orthodontic fixed appliance (at the end of the active phase of orthodontic treatment) in growing patients, due to unfavorable growth trends. Unfortunately, no prediction can be made regarding the risk of relapse in the event of giving up the retention phase [6]. So far, the specialized literature has described several types of retainers that can be classified as removable retainers and fixed retainers [7,8,9,10]. There is no unanimous consensus for any of the existing types of retainers, each with advantages and disadvantages [11,12].

The Hawley retainer (HR) and the vacuum-formed retainer (VFR) are the most common removable retainers. The Hawley retainer was introduced 100 years ago by Charles Hawley (1919) and has been widely used over time. It consists of an acrylic mass applied to the palatal mucosa, in contact with the palatal surfaces of all the teeth on the dental arch, Stahl or Adams clasps, and a labial bow made of stainless-steel wire. The labial bow passes from 4 or 6 anterior teeth and its advantages can include controlling the incisor torque and allowing vertical movement of posterior teeth [11]. The main disadvantages of the HR are related to the large extension on the palate and the visibility of the labial bow. VFR is considered an alternative to HR and is also known as clear thermoplastic retainers. Sheridan et al. [13] introduced a removable retainer in 1993, a vacuum-formed retainer that has been extensively used in the last years under the commercial name of Essix (DENTSPLY Raintree Essix Glenroe, Sarasota, FL, USA). This retainer is made from polyvinyl siloxane sheets to cover all the surfaces of the teeth [14].

Other materials used for the thermoplastic retainers are the following: polypropylene polymer-based material (Invisacryl C, Essix C+), polyethylene copolymers (Essix ACE), and polyethylene terephthalate glycol copolymer—PETG (a hard sheet material) [14,15]. The advantage of polyethylene polymers is that they allow the acrylic to be bonded to them. Therefore, polyethylene polymers are preferred in cases where bite planes must be incorporated into the appliances. Polypropylene polymers are more durable and flexible, but esthetically speaking, they are inferior to polyethylene as they are translucent. Moreover, acrylic cannot be added to the polypropylene polymers [16].

The advantages of these VFRs include superior esthetics, low cost, and ease of fabrication. The most common disadvantages are fractures (poor wear resistance and durability), occlusal wear, and limited vertical settling of teeth [14,17,18]. Frequently, the thicknesses of VFRs are 0.75 mm (0.30 inch) or 1 mm (0.40 inch) [19,20]. Over the last decades, many studies have been published on the effectiveness, advantages, disadvantages of retainers, patient comfort (patient satisfaction, accommodation period, compliance, speech, mastication) [21,22,23], periodontal health [5], and the costs involved in the retention stage [8], but all these studies revealed much controversy about the risks of relapse.

The aim of our study was to comparatively analyze the behavior of two types of removable retainers—HR and VFR—in terms of retainer damage (fracture, discoloration), loss, and the rate of installation of mild or severe relapse, requiring recourse to certain therapeutic interventions.

## 2. Materials and Methods

### 2.1. Patients

A retrospective study was performed on 900 orthodontic patients aged 11–17.9 years, randomly selected, for whom removable retainers—HRs and VFRs—were used on the upper arch during the retention phase. Of these, 469 patients (52.11%) were girls, and 431 patients (47.89%) were boys. The study was conducted in accordance with the World Medical Association (WMA) Declaration of Helsinki—Ethical Principles for Medical Research Involving Human Subjects, approved by the Ethics Committee of the University of Oradea, Romania (Project identification code: 10/15.10.2020). All patients were included in the study with their parents or legal tutors’ consent. All subjects gave their informed consent for inclusion before they participated in the study. The inclusion criteria in this retrospective study were as following: (1) patients to whom the phase of active orthodontic treatment at the upper arch was performed with fixed orthodontic appliances; (2) patients to whom a correct dental alignment/correct three-dimensional positioning was achieved, and inclination within normal limits of the upper frontal group; (3) patients who could be monitored for a period of at least 1 year in the retention phase; (4) crowding at the start of the orthodontic treatment, good alignment and clinically acceptable outcome at the end of the active treatment (zero values of the irregularity index for the 6 anterior superior teeth); (5) no systemic or oral disease. The exclusion criteria were the following: (1) patients who did not present for consultation for a period of at least one year in the retention phase; (2) patients who could not achieve the correct three-dimensional alignment and positioning of the teeth; (3) patients with cleft lip or palate, sectional fixed orthodontic treatment in the upper arch, severe rotations or midline diastema suggesting the need for a bonded retainer, poor periodontal condition; (4) patients with a restorative need in the labial segment (e.g., implant, bridges, veneers) or missing teeth.

Following the application of the inclusion/exclusion criteria, 618 patients remained in the study, out of which 352 were patients with vacuum-formed retainers (VFRs group). Out of these 352 patients with VFRs, 181 patients (51.42%) were girls and 171 patients (48.57%) were boys. The remaining 266 patients wore Hawley retainers (HRs group), of these 137 patients (51.50%) were girls and 129 patients (48.49%) were boys. Patients in both groups were advised to wear the retainers full-time for 3–4 months (except during meals) and then night-only. We performed an analysis of the two batches at 6 months (T1) and 12 months (T2) after the application of the retainer.

### 2.2. Measurements

The assessment of the correct teeth positioning at the end of the active orthodontic treatment was made by clinical observation, measuring irregularity index using study models (Figure 1a), and by cephalometric assessment of the inclination of the upper incisors. Thus, we assessed the anterior–posterior position of the upper incisors at the end of the orthodontic treatment by measuring 1—A-Pog (upper central incisor to A-Pog) angle (angle formed by the extension of the longitudinal axis of maxillary incisor to the A-Pog line, with normal values of 3 ± 5 degrees) on the lateral cephalograms. To measure the 1—A-Pog values, we used a computerized defalcation software entitled OnyxCeph version 69 (open software license) (Figure 1b). In order to analyze the correct alignment of the anterior maxillary teeth at the end of the orthodontic active treatment (prior to the application of the removable retainer), but also during the first year of the retention phase in cases with changes in the position of the upper teeth (respectively the cases with relapse), irregularity indexes (similar to the index introduced by Little R.M. [24] for the six anterior mandibular teeth—Little’s irregularity indexes) were measured on dental casts. The irregularity index was calculated based on the linear measurement of displacements in the anatomical contact points of upper anterior teeth, parallel to the occlusal plane (Figure 1a). We used a digital caliper with a 0.01 mm sensitivity for these measurements. All cephalometric analyses and dental cast measurements were performed by the same investigator to avoid inter-operator bias.

### 2.3. Procedure

The Hawley retainers had Adams clasps attached, labial bow with vertical loops, and palatal acrylic. The acrylic part of the Hawley retainer had a uniform thickness (2–3 mm), and it was trimmed into a horseshoe shape. The labial bow extended to the posterior region (molar-to-molar) and was soldered to the Adams clasps. It can be effective in controlling incisor torque and the position of the canines and premolars as well (Figure 2).

The VFRs were manufactured from polyethylene terephthalate glycol copolymer (PETG) 1 mm thickness (0.040″) sheet materials according to the manufacturer’s instructions, and these maxillary retainers were trimmed into a horseshoe shape. To obtain the VFRs, we used a vacuum machine that adapts heat-softened plastic by negative pressure, creating a vacuum, and pulls the plastic onto a working cast (Figure 3).

The efficiency of both types of removable retainers was evaluated according to the following criteria:-Damage of the retainers (fractures—perforated and cracked, loss, unfitting and obvious discolorations) as a result of which the respective retainers could no longer be used, requiring restoration.-Loss of the retainer.-The rate of installation of mild or severe relapse of dento-maxillary anomalies, requiring the use of various therapeutic interventions.

We considered as mild relapse situations where the irregularity index had values of maximum 1 mm (these cases can be solved with the same Hawley retainer or with a new VFR on the set-up model) or changes in the inclination of the upper incisors (1 to A-Pog) were maximum 2 degrees from the value at the end of the active phase of the orthodontic treatment. Cases with mild relapse can be solved by activating the Hawley retainer or by manufacturing a thermoplastic retainer on the setup model. We considered as severe relapse the cases where the irregularity index was over 1 mm or with 1 to A-Pog value differences of over 2 degrees compared to the value at the end of the active phase of the orthodontic treatment; these situations required the reapplication of fixed orthodontic appliances or the use of clear aligners.

### 2.4. Data Analysis

All the data were analyzed using IBM SPSS Statistics 25. Quantitative variables were expressed as averages with standard deviations and medians with interquartile ranges. Shapiro–Wilk test was used for distribution testing. Qualitative variables were expressed as counts with percentages and were tested using Fisher’s Exact Test for evaluating statistical differences between groups. Quantitative variables with non-parametric distribution were tested using the Mann–Whitney U test.

## 3. Results

Data presented in Table 1 show the characteristics of the studied patients. The average age was 13.98 ± 1.51 years with a median age of 13.7 years. VFRs were applied to 57% of the patients and HRs to 43% of them. Six percent of all the retainers were damaged, mostly at T2 (54.1%), while 7% of all retainers were lost, mostly at T1 (58.1%). Of all the patients, 9.1% presented mild relapse, mostly in the first 6 months (58.9%), while 2.6% presented severe relapse, mostly in the first 6 months (62.5%).

Table 2 and Figure 4 present a comparison of patients’ age according to the type of retainer applied. The distribution of the patients’ age was detected to be non-parametric according to the Shapiro–Wilk test (*p* < 0.001). According to the Mann–Whitney U test, the differences in the patients’ age according to the type of retainers was not significant (*p* = 0.983), and the median age in both groups was 13.7 years.

The distribution of the patients according to the occurrence of damage and the type of retainer is presented in Table 3 and Figure 5. Differences between groups were detected as significant according to Fisher’s Exact Test (*p* < 0.001), emphasizing that the VFRs are significantly more frequently associated with the existence of damage compared with the HRs. The odds ratio is 5.22 (95% C.I: 2.006–13.587), which shows that it is 5.22 times more probable that VFRs would be damaged.

Data from Table 4 and Figure 6 show the distribution of the patients according to the type of damage and type of retainer. Differences between groups were detected as not significant according to Fisher’s Exact Test (*p* = 1.000), as no type of retainer was associated with some particular type of damage. However, it is worth mentioning that the damaged retainers were mostly fractured at T2.

Data from Table 5 and Figure 7 show the distribution of the patients according to the existence of loss and type of retainer. Differences between groups were detected as not significant according to Fisher’s Exact Test (*p* = 0.750), as such no type of retainer was associated with a higher frequency of loss.

Data from Table 6 and Figure 8 show the distribution of the patients according to the type of loss and type of retainer. Differences between groups were detected as not significant according to Fisher’s Exact Test (*p* = 1.000), as the retainers were not associated with any particular type of loss. However, it is worth mentioning that the lost retainers were mostly lost at T1.

Data from Table 7 and Figure 9 show the distribution of the patients according to the existence of mild relapse and type of retainer. Differences between groups were detected as not significant according to Fisher’s Exact Test (*p* = 0.480), as no particular type of retainer was associated with a higher frequency of a mild relapse.

Data from Table 8 and Figure 10 show the distribution of the patients according to the type of mild relapse and type of retainer. Differences between groups were detected as not significant according to Fisher’s Exact Test (*p* = 1.000), as the retainers were not associated with any particular type of mild relapse. However, it is worth mentioning that patients that had a mild relapse, usually had mild re-lapse at T1.

Data from Table 9 and Figure 11 show the distribution of the patients according to the existence of severe relapse and type of retainer. Differences between groups were detected as significant according to Fisher’s Exact Test (*p* = 0.042); data show that the HRs are significantly more frequently associated with the existence of severe relapse than the VFRs. The odds ratio 0.334 (95% C.I: 0.115–0.973) shows that it is 3 times more probable that HRs retainers would be associated with severe relapse.

Data from Table 10 and Figure 12 show the distribution of the patients according to the type of severe relapse and type of retainer. Differences between groups were detected as not significant according to Fisher’s Exact Test (*p* = 1.000), as such the retainers were not associated with any particular type of severe relapse. However, it is worth mentioning that patients who had a severe relapse usually had it in the first 6 months.

## 4. Discussion

The study we conducted shows that VFRs are less resistant than HRs, suffering much more fractures compared to HRs (*p* < 0.001) (Table 3). Mild relapses were reported in both groups without any statistical significance, being more frequently at T1 in both groups—due to inadequate wearing, damage, or loss of retainer. Mild relapses could be remediated by activating the HRs or by manufacturing the VFRs on the setup models. However, severe relapses were significantly more frequent in the HRs group (*p* < 0.05) (Table 9), especially due to improper wearing. These situations required the resumption of the active orthodontic treatment with a fixed appliance or clear aligners.

For the retention phase, we most often use removable retainers in the upper arch and fixed retainers in the lower arch [10,25]. Despite all the disadvantages of the need for proper patient compliance, we are most often forced to use removable retainers in the upper arch as fixed retainers can generate occlusive interference. In our practice, we used HRs in the upper arch for a long time, but due to dissatisfaction with the use of HRs, we decided to apply thermoplastic retainers. Thus, the use of VFRs as upper retainers has predominated in our practice for the last five years. Of course, there were various drawbacks in the use of VFRs, which led us to this retrospective study that analyzes the behavior of the two types of retainers in terms of certain clinical and paraclinical aspects. VFRs are considered to be less resistant in time, and their durability often is limited to a few months. However, Gardner et al. [14] demonstrated that polyethylene terephthalate glycol copolymer (PETG) is harder and has greater resistance to wear than Invisacryl C and Essix C+, which are softer, polypropylene-based thermoplastics. We also prefer to use PETG as removable retainer materials. In this study, the slight occlusal wear of the VFRs was not considered as damage because it did not affect the stability of the teeth position. In our opinion, the fact that a slight degree of wear is installed in the areas with more intense occlusal contact points allows a continuation of the settling phase, which has a beneficial effect on the occlusal stability. One of the important advantages of polyethylene polymers used for VFRs retainers is that it allows acrylic pads to be bonded to the material in the anterior region (near the upper lateral incisors) on the upper retainer and in the posterior region on the lower VFRs, which allows application in the retention phase of light class II elastics (or light class III elastics, with a reverse orientation) in patients who have used class II or class III mechanics and present the risk of relapse of mandibular-maxillary relationships [26,27]. We inform patients that VFRs may change their translucency after one year of use, color changes may occur, so it is often necessary to restore them after the first year of retention.

The literature presents various protocols for wearing these mobile retainers during the retention phase. Some authors recommend prolonging the retention phase for a period of one year, with full-time wearing in the first 3–4 months of retention, followed by night-only wearing for a period of up to one year [28], or full-time wearing of VFRs in the first week, followed by night-only wearing [29], while other authors have demonstrated that full-time wearing is not beneficial compared to night-only wearing [11]. Shawesh et al. [30] conducted a study starting from the idea that it is still unclear whether it may be clinically acceptable for patients to wear their retainers for 1 year at night only or whether it is necessary for an initial period of full-time wear followed by night-only wear. Their study demonstrated that there were no statistically significant differences between the two retention protocols (full-time versus night-only wear) for labial segment irregularity or crowding and thus the patients may be advised to wear their retainers at night only. On the other hand, in a study conducted by Meade and Millett (2014) on a sample of orthodontic specialists, most of the respondents from their study (70–76%) preferred a full-time followed by part-time wear protocol of removable retainers, and almost all respondents (93%) agreed to full occlusal coverage design. The authors found that VFRs were the most chosen retainers, recommended by 53% of the respondents in the maxilla and 33% in the mandible [19]. In our practice in adult patients, we recommend that the removable retainers be worn two years after the completion of the active phase of the orthodontic treatment. In contrast, in children and adolescent patients at the end of the active orthodontic treatment, we recommend that retention be continued until the end of growth. We instructed our patients to wear their retainers full-time (except during meals) for a period of 3–4 months and then only while sleeping. The retention protocol we use provides a gradual reduction in wear time in the last 4–6 months of the retention period (alternative wearing, one night yes–one night no, then wearing only two nights/week, and finally wearing only one night/week). In this way, the risk of relapse can be investigated, signaled by the fact that the removable retainer begins to fit perfectly no longer on the teeth.

Other authors did not find statistically significant differences between the two types of retainers. Kaya Y. et al. (2019) did not obtain statistically significant differences between two groups of patients who used the Hawley retainer and the Essix retainer for a period of 12 months in terms of overbite, overjet, arch length, maxillary, and mandibular intercanine widths, and intermolar widths [2]. In a systematic review, the authors found that, in terms of occlusal contacts and survival time, there was insufficient evidence to distinguish the effectiveness between HRs and VFRs. Some authors reported higher numbers of occlusal contacts for patients wearing HRs and higher patient satisfaction associated with VFRs [12]. It is well-known that the best orthodontic results are related to good occlusal contacts (centric stops) and intercuspation, and hence, various studies in the literature have reported changes in the occlusion, respectively increased the number of occlusal contacts after orthodontic treatment with conventional retention devices [31,32]. However, different results were obtained by Dinçer and Aslan [33], as the expected increase of occlusal contacts was not observed at the end of the retention period when using thermoplastic retainers. The authors also concluded that both ideal and non-ideal posterior contacts increased in the long term, but the number of non-ideal contacts was greater than the ideal contacts. In a study performed by Tecco et al. [34], the authors showed that upper removable retainers, unlike positioners, did not have a relevant effect on the activity of masticatory and neck muscles.

Other studies have shown that Essix retainers are more effective than Hawley retainers only in the mandibular arch, preventing the recurrence of crowding to a greater extent [25,35]. In another comparative study on Hawley and vacuum-formed retainers, performed by Barlin et al. (2011) [9], there were no statistically significant differences between the two types of retainers, and the authors suggested that the degree of relapse is unlikely to be affected by the choice of retainer. On the contrary, our study compared the two types of retainers only in the upper arch and it demonstrated that HR was associated with an increased risk of severe relapse in the first year of the retention phase. Barlin et al. [9] suggested that, when deciding on the type of retainer to be fitted following fixed appliance therapy, other factors such as cost may play a more significant role when deciding what type of retainer should be used. In our praxis, the costs with HRs are double compared to those with VFRs.

Most of the patients (8 patients) with severe relapse in our study, belonging to the HRs group, initially showed severe canine abnormalities (accentuated ectopia or impactions) [36] that suggests that HR are associated with a higher risk of relapse in patients who have been orthodontically treated for severe canine malposition. Two of the patients in the VFRs group with severe relapse initially presented canine abnormalities associated with severe dental crowding, while three patients initially presented severe protrusions.

The removable retainers are efficient in the orthodontic practice in terms of maintaining the stability of the result of the active orthodontic treatment. They have many advantages related to the possibility of maintaining proper oral hygiene [37], are easy to perform [38], are acceptable in terms of aesthetics (especially the VFRs), and have a satisfactory resistance [39]. Their main disadvantage is related to the need for optimal collaboration of the patient. Many children and adolescents in our study did not present at regular check-ups during the retention phase and did not exactly follow our recommendations. Therefore, like other authors, we consider it particularly important to have optimal communication with pediatric patients [40]. In the current context of young patients’ education and their participation in social media activity, several studies have pointed out various aspects related to patient compliance and have suggested strategies to obtain the best patient engagement in order to reduce the occurrence and degree of relapse [41]. The use of social networks or regular communication, written or through shots, with patients on WhatsApp has a beneficial role in stimulating patient compliance in terms of increasing the quality of oral hygiene, increasing regularity in wearing removable retainers, participation in the follow-up program to achieve better long-term results of the orthodontic stability [41,42]. This study and others that we have carried out previously lead us toward giving special importance to stimulating patient compliance [43], training patients on how to store and properly maintain these removable retainers, trying to always make them aware of the fact that non-compliance retention favors relapse, the treatment instituted being thus in vain. However, given the increased prevalence of malocclusions, their early onset often in early childhood, involving several factors (e.g., dental caries and its complications, vicious oral habits), the high risk of relapse of malocclusions and that many children and adolescents are not aware of the importance of the retention phase, we consider it very important to implement a prevention program in pedodontics and orthodontics starting with pre-school children, program that requires close collaboration between parents, educators, and the professional team—pedodontists, orthodontists, and pediatricians [44,45].

The limitations of the study are primarily related to the inconsistency of the information provided by children and adolescents on the regular wearing of retainers, with the information provided being often irrelevant. Relapse in the upper arch also involves a complex set of factors such as gingival fibers, periodontal tissues, oral habits, occlusion, lower arch relapse, facial soft tissues, and the impossibility of establishing exactly the factors that contributed to its occurrence. Our study was limited to analyzing the behavior of the two types of retainers and the relapse rate over a limited retention period (one year), but the retention phase often extends over a longer period, sometimes life-long retention is necessary. Therefore, further studies are needed, on a larger sample size, with the investigation of the various parameters involved in relapse, over an extended period.

## 5. Conclusions

Following the study and being aware that there is no perfect device that can be used as a retainer in orthodontic practice, we can conclude that the two types of removable retainers, namely, the Hawley retainer and the thermoplastic retainer, are effective in the upper arch in the retention phase. However, the thermoplastic retainer has lower resistance in time compared to the Hawley retainer. No differences between groups were found in the rates of loss of the retainers. The risk of severe relapse during the first year of the retention phase is higher when using Hawley retainers.

## Figures and Tables

**Figure 1 children-07-00295-f001:**
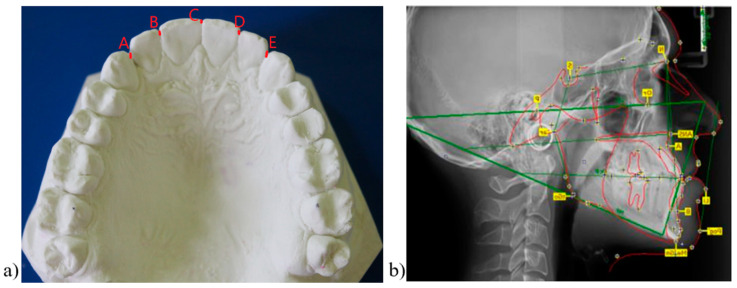
(**a**) Dental cast measurements: Irregularity index (A + B + C + D + E); (**b**) Cephalometric analysis using OnixCeph software.

**Figure 2 children-07-00295-f002:**
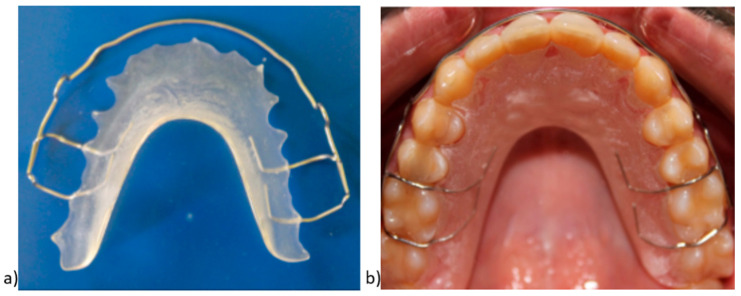
(**a**) Hawley retainer (HR); (**b**) HR view in the oral cavity.

**Figure 3 children-07-00295-f003:**
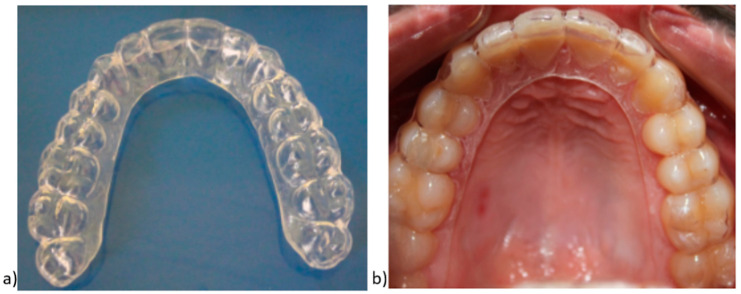
(**a**) Vacuum-formed retainer (VFR); (**b**) VFR view in the oral cavity.

**Figure 4 children-07-00295-f004:**
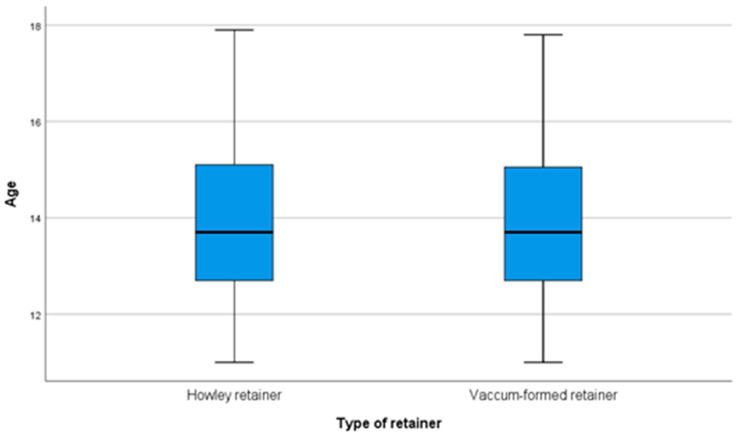
Comparison of patients’ age according to the type of retainer.

**Figure 5 children-07-00295-f005:**
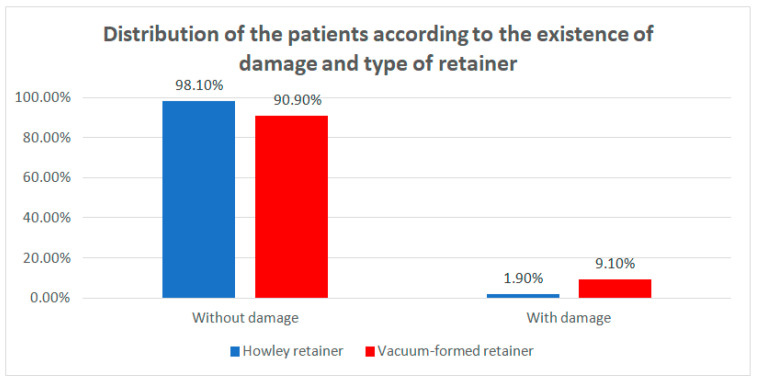
Distribution of the patients according to the existence of damage and type of retainer.

**Figure 6 children-07-00295-f006:**
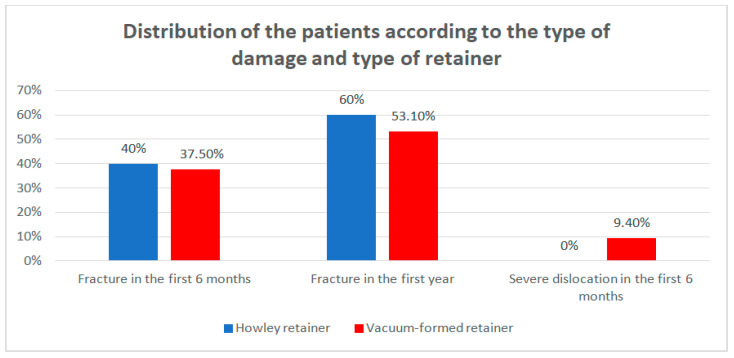
Distribution of the patients according to the type of damage and type of retainer.

**Figure 7 children-07-00295-f007:**
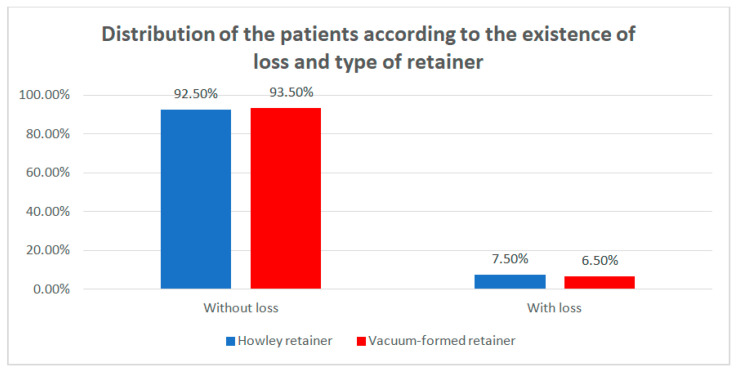
Distribution of the patients according to the existence of loss and type of retainer.

**Figure 8 children-07-00295-f008:**
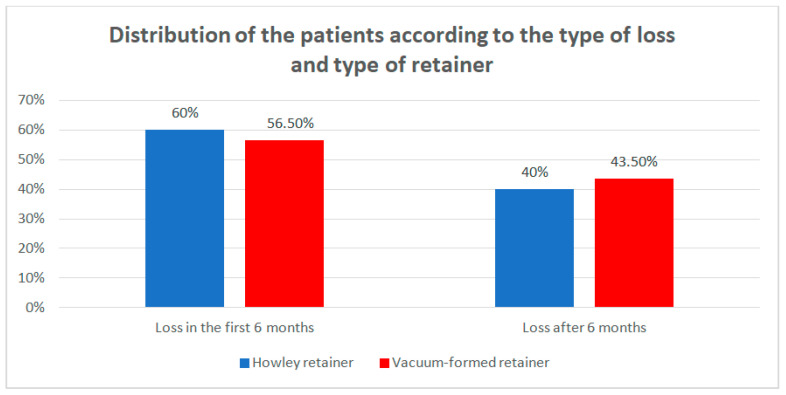
Distribution of the patients according to the type of loss and type of retainer.

**Figure 9 children-07-00295-f009:**
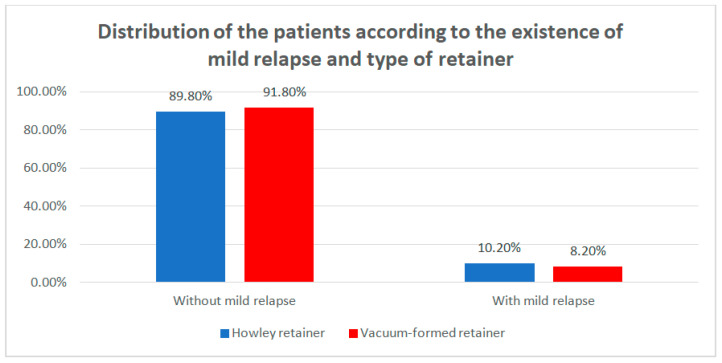
Distribution of the patients according to the existence of mild relapse and type of retainer.

**Figure 10 children-07-00295-f010:**
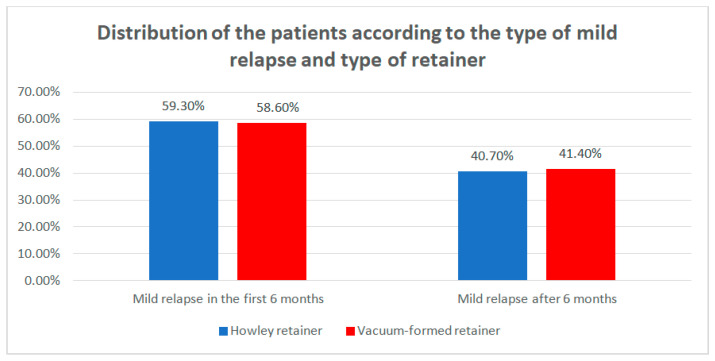
Distribution of the patients according to the type of mild relapse and type of retainer.

**Figure 11 children-07-00295-f011:**
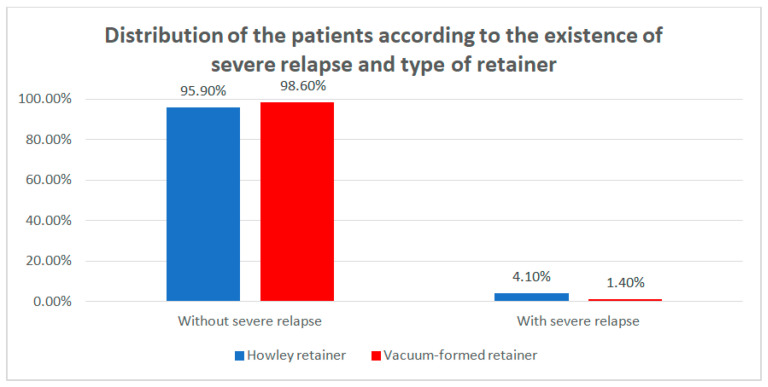
Distribution of the patients according to the existence of severe relapse and type of retainer.

**Figure 12 children-07-00295-f012:**
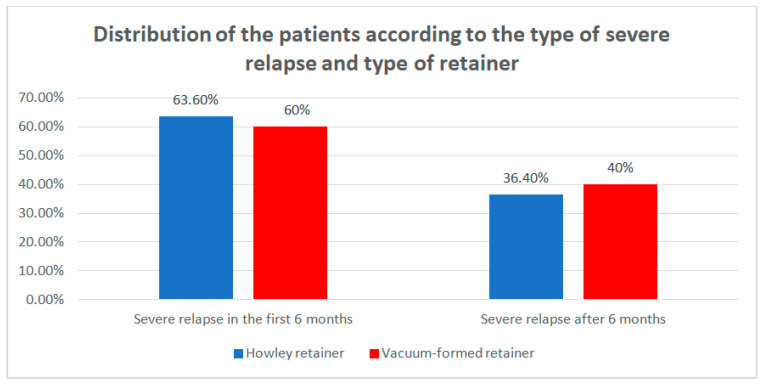
Distribution of the patients according to the type of severe relapse and type of retainer.

**Table 1 children-07-00295-t001:** Characteristics of the studied patients.

Parameter	Value
Age (Average ± SD, Median–IQR)	13.98 ± 1.51, 13.7 (12.7–15.1)
Retainer type (No./%)	266 (43%) HR/352 (57%) VFR
Damage (No./%)	37 (6%) Damaged retainers
Type of damage (No./%)	
Fracture at T1	14 (37.8%)
Fracture at T2	20 (54.1%)
Severe discoloration at T1	3 (8.1%)
Loss (No./%)	43 (7%) Lost retainers
Type of loss (No./%)	
Loss at T1	25 (58.1%)
Loss at T2	18 (41.9%)
Mild relapse (No./%)	56 (9.1%)
Type of mild relapse (No./%)	
Mild relapse in the first 6 months	33 (58.9%)
Mild relapse after 6 months	23 (41.1%)
Severe relapse (No./%)	16 (2.6%)
Type of severe relapse (No./%)	
Severe relapse in the first 6 months	10 (62.5%)
Severe relapse after 6 months	6 (37.5%)

HR: The Hawley retainer; VFR: the vacuum-formed retainer; SD: standard deviation; IQR: interquartile ranges.

**Table 2 children-07-00295-t002:** Comparison of patients’ age according to the type of retainer.

Retainer	Average ± SD	Median (IQR)	Average Rank	*p* *
HR (*p* < 0.001 **)	13.98 ± 1.53	13.7 (12.7–15.1)	309.33	0.983
VFR (*p* < 0.001 **)	13.97 ± 1.49	13.7 (12.7–15.07)	309.63	

* Mann–Whitney U Test, ** Shapiro–Wilk Test.

**Table 3 children-07-00295-t003:** Distribution of the patients according to the occurrence of damage and type of retainer.

Type of Retainer/Damage	Hawley Retainer		Vacuum-Formed Retainer		*p* *
	No	%	No	%	
No damage	261	98.1%	320	90.9%	<0.001
Damage	5	1.9%	32	9.1%	

* Fisher’s Exact Test.

**Table 4 children-07-00295-t004:** Distribution of the patients according to the type of damage and type of retainer.

Type of Retainer/Damage	Hawley Retainer		Vacuum-Formed Retainer		*p* *
	No	%	No	%	
Fracture at T1	2	4%	12	37.5%	1.000
Fracture at T2	3	60%	17	53.1%	
Severe discoloration at T1	0	0%	3	9.4%	

* Fisher’s Exact Test. T1: after 6 months of retention; T2: after 12 months of retention.

**Table 5 children-07-00295-t005:** Distribution of the patients according to the existence of loss and type of retainer.

Type of Retainer/Loss	Hawley Retainer		Vacuum-Formed Retainer		*p* *
	No	%	No	%	
Without loss	246	92.5%	329	93.5%	0.750
With loss	20	7.5%	23	6.5%	

* Fisher’s Exact Test.

**Table 6 children-07-00295-t006:** Distribution of the patients according to the type of loss and type of retainer.

Type of Retainer/Loss	Hawley Retainer		Vacuum-Formed Retainer		*p* *
	No	%	No	%	
Loss at T1	12	60%	13	56.5%	1.000
Loss at T2	8	40%	10	43.5%	

* Fisher’s Exact Test.

**Table 7 children-07-00295-t007:** Distribution of the patients according to the existence of mild relapse and type of retainer.

Type of Retainer/Mild Relapse	Hawley Retainer		Vacuum-Formed Retainer		*p* *
	No	%	No	%	
Without mild relapse	239	89.8%	323	91.8%	0.480
With mild relapse	27	10.2%	29	8.2%	

* Fisher’s Exact Test.

**Table 8 children-07-00295-t008:** Distribution of the patients according to the type of mild relapse and type of retainer.

Type of Retainer/Mild Relapse	Hawley Retainer		Vacuum-Formed Retainer		*p* *
	No	%	No	%	
Mild relapse at T1	16	59.3%	17	58.6%	1.000
Mild relapse at T2	11	40.7%	12	41.4%	

* Fisher’s Exact Test.

**Table 9 children-07-00295-t009:** Distribution of the patients according to the existence of severe relapse and type of retainer.

Type of Retainer Severe Relapse	Hawley Retainer		Vacuum-Formed Retainer		*p* *
	No	%	No	%	
Without severe relapse	255	95.9%	347	98.6%	0.042
With severe relapse	11	4.1%	5	1.4%	

* Fisher’s Exact Test.

**Table 10 children-07-00295-t010:** Distribution of the patients according to the type of severe relapse and type of retainer.

Type of Retainer/Severe Relapse	Hawley Retainer		Vacuum-Formed Retainer		*p* *
	No	%	No	%	
Severe relapse at T1	7	63.6%	3	60%	1.000
Severe relapse at T2	4	36.4%	2	40%	

* Fisher’s Exact Test.

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
