# Peer review of "The Behavior of Two Types of Upper Removable Retainers—Our Clinical Experience"

_children, 2020, doi:10.3390/children7120295_

Round 1
Reviewer 1 Report
more information in the selection criteria potential variation between groups and bias in selection of patients should be provided.
Author Response
Dear reviewer,
Please see the attachment.
Yours truly,
Ligia Vaida

Reviewer 2 Report
I reviewed the manuscript: The Behavior of Two Types of Upper Removable 2 Retainers – Our Clinical Experience. Thank you for asking me to review this paper. In my opinion the manuscript is not well-structured and the topic is not particularly interesting. I think that this paper does not make a contribution to new knowledge in the discipline or the application of knowledge.
Author Response
Dear Reviewer,
Please see the attachment.
Yours truly,
Ligia Vaida

Reviewer 3 Report
Dear Authors, the work is original and really interesting.
Some improvement shall be useful.
In the patients' selection, you decided to evaluate upper arch relapse. please, could you clarify and better explain this decision? What about the treatment and the relapse in the lower arch? The treatment of the lower arch could affect the stability of the occlusion and lower teeth could be subject to relapse too.
It might be useful to clarify, in the discussion section, how the damage of VFRs occurs and whether the wearuing of occlusal surfaces of VFR could affect the stability of teeth position. What do you think about this phenomenon in the lower arch?
In the discussion section (lines 326-329), it is more opportune to cite literature approaching the relapse and compliance issues. The following study could be really useful:
Zotti F, Zotti R, Albanese M, Nocini PF, Paganelli C. Implementing post-orthodontic compliance among adolescents wearing removable retainers through Whatsapp: a pilot study. Patient Prefer Adherence. 2019 Apr 23;13:609-615. doi: 10.2147/PPA.S200822. PMID: 31118585; PMCID: PMC6498955.
Furthermore, what do you think about the occlusal interferences of VFR. In your opinion, could VFR influence occlusal stability and muscle activity? Please, provide some references and discuss them in the discussion section.
Author Response

(The authors gave the same response as above.)
